# SASMOTE: A Self-Attention Oversampling Method for Imbalanced CSI Fingerprints in Indoor Positioning Systems

**DOI:** 10.3390/s22155677

**Published:** 2022-07-29

**Authors:** Ankang Liu, Lingfei Cheng, Changdong Yu

**Affiliations:** 1School of Physics and Electronic Information Engineering, Henan Polytechnic University, Jiaozuo 454000, China; 212011010016@home.hpu.edu.cn; 2College of Information and Communication Engineering, Harbin Engineering University, Harbin 150001, China; ycd_darren@163.com

**Keywords:** indoor localization, channel state information (CSI), fingerprinting, imbalanced data, deep learning, self attention

## Abstract

WiFi localization based on channel state information (CSI) fingerprints has become the mainstream method for indoor positioning due to the widespread deployment of WiFi networks, in which fingerprint database building is critical. However, issues, such as insufficient samples or missing data in the collection fingerprint database, result in unbalanced training data for the localization system during the construction of the CSI fingerprint database. To address the above issue, we propose a deep learning-based oversampling method, called Self-Attention Synthetic Minority Oversampling Technique (SASMOTE), for complementing the fingerprint database to improve localization accuracy. Specifically, a novel self-attention encoder-decoder is firstly designed to compress the original data dimensionality and extract rich features. The synthetic minority oversampling technique (SMOTE) is adopted to oversample minority class data to achieve data balance. In addition, we also construct the corresponding CSI fingerprinting dataset to train the model. Finally, extensive experiments are performed on different data to verify the performance of the proposed method. The results show that our SASMOTE method can effectively solve the data imbalance problem. Meanwhile, the improved location model, 1D-MobileNet, is tested on the balanced fingerprint database to further verify the excellent performance of our proposed methods.

## 1. Introduction

The localization of mobile devices is becoming increasingly significant, and indoor WiFi localization has become a hot research area, thanks to the rapid expansion of WiFi networks and the explosive rise of location-based services, such as indoor navigation, indoor tracking, and activity recognition [1,2,3]. Various approaches have been proposed successively to solve the indoor localization problems [4,5,6]. Some of these solutions necessitate the use of many access points (APs), and their effectiveness is heavily dependent on measurement accuracy, which is difficult to achieve with standard WiFi equipment. In contrast, indoor positioning technique based on fingerprint recognition has become a popular positioning technique because of its high precision and minimal hardware requirements [7,8,9]. The working principle of fingerprint-based localization methods is to extract feature information from the measured position, which is called “fingerprint”. The WiFi signal of the device to be located is then compared to the known feature in the real-time location phase to determine the location. The fingerprint-based localization approaches are typically divided into two stages: offline training stage and online localization stage. In the offline fingerprint training stage, the WiFi signals of each confirmed reference point are measured and collected as training samples for the localization model [7]. During the online localization stage, the WiFi signal of a specific reference point is fed to the trained localization model to predict the location results.

The received signal strength (RSS) and channel state information (CSI) techniques, as the two most commonly used fingerprints, have been widely used in previous WLAN-based indoor localization studies [10,11]. Although RSS is a fingerprint that can be easily measured at WiFi receivers, it has some unavoidable limitations. RSS is coarse channel data, which means it is vulnerable to dynamic environments that do not adequately reflect the relationship between channel characteristics and location. Meanwhile, RSS is challenging to implement across various mobile devices. For example, the RSS of a laptop and a cell phone device at the same location is different because they have different antenna characteristics. Instead, CSI uses varying signal intensities and phases in different subcarriers to deliver more precise multipath information than RSS [12]. Therefore, we choose to use the more reliable and mature CSI data as the fingerprint data for localization in this research.

However, one of the most significant drawbacks of fingerprint-based localization is the vast amount of data that must be assessed in the offline stage in order to ensure the correctness of the system [13]. Therefore, data collection work (i.e., building a fingerprint database) is time consuming and labor intensive. In fact, we cannot guarantee that the same number of fingerprints will be obtained at each reference location when collecting data. Meanwhile, some outliers are removed from the fingerprint data during processing, which will further lead to the imbalance of the fingerprint data. The fingerprint data imbalance problem in the database will affect the popularity and application of fingerprint-based location technology. In order to address the above issue, some approaches have been proposed successively [14,15,16,17]. Gu et al. [15] proposed a compressed sensing-based approach to recover absent fingerprints. However, this method only restores the missing data and does not resolve the imbalance problem between the fingerprint data. Based on the classic SMOTE algorithm [18], methods [16,17] utilize machine learning methods to solve the problem of data imbalance between and within classes. Nevertheless, these methods do not solve the problem of ambiguity of one-dimensional long sequence data, and the generated data are easily affected by noise. In addition, it is worth noting that there is no deep learning method to solve the imbalance problem of fingerprint data. It is interesting to introduce deep learning techniques to this research topic.

Inspired by the above principles, we put forward a novel approach in this article, called the Self Attention Synthetic Minority Oversampling Technique (SASMOTE), to effectively solve the imbalance problem of fingerprint data. The specific composition of the entire framework is as follows: a self-attention encoder firstly extracts features from fingerprint data. Then, an oversampling method based on the SMOTE method is adopted to populate minority class data samples. After that, a corresponding decoder is used to output the balanced fingerprint data. In addition, we also specially built the corresponding CSI fingerprint database for model training and testing. Finally, we perform extensive experiments to evaluate the performance of the model. We use the improved 1D-MobileNet to test the balanced fingerprint database to further verify the performance of the proposed framework. In summary, the main contributions of this work are summarized as follows:We propose a deep learning-based oversampling method to end-to-end deal with the imbalanced fingerprint database. In addition, a corresponding fingerprint dataset is collected and constructed for model training and testing. To the best of our knowledge, we are the first to study the problem of constructing a fingerprint database that encounters data imbalance.In the framework, we design a self-attention encoder-decoder to extract and integrate data features. Meanwhile, the SMOTE algorithm is integrated into the encoder-decoder to supplement the small number of sample data, which solves the problem of fuzzy features in high-dimensional data.Extensive experiments are conducted in real environments and the results show that the proposed method has better performance compared to existing oversampling methods. The new fingerprint library generated by SASMOTE is applicable to other localization methods, such as the 1D-MobileNet model.

The rest of the article is structured as follows. The Section 2 reviews the related work. The core idea of this study is briefly described in Section 3.1. The backdrop of CSI and the workflow of the localization system are presented in Section 3.2 and Section 3.3. Our proposed SASMOTE method, including training algorithms and evaluation measures, is introduced in Section 4. Section 5 reports the simulation and experimental results. Finally, the conclusions and discussions are presented in Section 6.

## 2. Related Works

Although it is quite reliable to establish fingerprint databases for localization, collecting training samples to create fingerprint databases usually requires enormous considerable manpower. To lessen the human workload, many researchers have developed efficient methods to complete the fingerprint databases [15,19]. As mentioned above, the problem of data imbalance between classes inevitably occurs in the construction of the fingerprint database. This has a great impact on the classification accuracy of the later localization model. At present, there is also a lot of work devoted to solving the data imbalance problem. Data imbalance is a widespread research problem in the field of engineering technology. Generally, there are three strategies used to solve this issue: (1) Reduce the number of majority classes to balance the data, which is called undersampling [20]; (2) Increase the number of minority classes to achieve data balance, which is called oversampling [18]; (3) Reduce the number of the majority class while increasing the number of minority class, which can be called a hybrid approach [21]. Traditional undersampling and oversampling algorithms can be performed with low computational complexity, but this can lead to instability in new datasets (e.g., removing important instances or adding noisy instances in the process). Therefore, intelligently selecting instances to preprocess imbalanced data is a popular research direction. In this section, we mainly review these related works, which can be divided into traditional machine learning based methods and deep learning based methods.

### 2.1. Machine Learning Based Methods

Machine learning, as an important branch of artificial intelligence, is widely used in various research fields. Similarly, there are also some classical machine learning based methods to solve the problem of imbalanced data distribution. For instance, Hoyos-Osorio et al. [22] propose a Relevant Information-Based Undersampling (RIUS) method to select the most relevant instances from the majority class to enhance classification performance on imbalanced data. There are few solutions for directional undersampling, whereas the oversampling technique has obtained greater attention as a result of the popularity of the SMOTE approach [18]. The SMOTE algorithm is a classic machine learning-based oversampling method. It generates new synthetic samples from the k-nearest neighbors (KNN) of the minority samples. Compared with the simple traditional oversampling method, the dataset generated by the SMOTE method has strong generalization. In this way, the problem of overfitting the model caused by the simple sampling method can be effectively overcome. Based on the SMOTE algorithm, different excellent variants have been proposed successively [16,17,23]. Lee et al. [16] present an approach by combining Gaussian probability distribution and the SMOTE algorithm. The results show that their method can solve the class imbalance problem. Douzas et al. [17] propose an effective oversampling method based on k-means clustering and the SMOTE algorithm, which avoids the generation of noise and overcomes imbalances between and within classes. Nevertheless, these methods are essentially based on the principle of KNN and cannot solve the problem of fuzzy characteristics of high-dimensional data, such as fingerprint data. Aiming at the class-imbalance problem in multi-label datasets, Mishra et al. [24] introduce a feature construction method based on the SMOTE approach. It takes the distances of the minority class to all instances as features, and then uses the SMOTE algorithm to balance the ratio between minority and majority instances. However, the accuracy of the method is greatly affected when outliers appear in the dataset. In the latest work [25], Yi et al. propose a simple and effective method called ASNSMOTE, which filters the noise in the minority class by determining whether the nearest neighbor of each minority instance belongs to the minority class or the majority class. However, it only considers the noise problem of the boundary.

### 2.2. Deep Learning Based Methods

Deep learning technology, as the most important branch of machine learning, has achieved great success in various fields because of its powerful feature representation ability [26,27,28]. Compared with machine learning, deep learning models can handle mass and complicated data, which is required for processing complicated fingerprint data. Generative Adversarial Networks (GANs) [29] are attractive to researchers because they can perform techniques similar to oversampling to generate the desired data. Douzas et al. [27] introduce a network based on conditional generative adversarial networks (CGAN), which generates minority class samples by setting up known conditional information to the GAN model. Nonetheless, the model may crash due to a lack of sample size to support it [30]. Li et al. [31] collect CSI data into amplitude feature maps and extend the fingerprint database using the proposed Amplitude-Feature Deep Convolutional Generative Adversarial Network (AF-DCGAN) model. However, GAN models are good at processing image data and are not so effective at processing one-dimensional input data (e.g., CSI). Therefore, it is a challenge to efficiently process one-dimensional fingerprint data using deep learning methods. We propose a novel self-attention encoder-decoder combined with SMOTE algorithm to handle the imbalanced fingerprint database in an end-to-end manner. Next, we will detail the motivation and rationale for building the framework.

## 3. Background and Framework

### 3.1. Basic Ideas

In the fingerprint-based indoor positioning system, the greater the imbalance of the fingerprint database data, the greater the impact on the positioning accuracy. We conducted relevant experiments to demonstrate this relationship. Specifically, the CSI data was collected in a classroom to create a fingerprint database. The schematic diagram of the environment plane of the localization experiments is shown in Figure 1. We adopt a TL-WR745 N wireless router as the transmitter that is equipped with one transmit antenna (AP in the figure). Additionally, a ThinkPad X201 laptop equipped with Inter Wireless Link 5300 network interface cards (NICs) and three antennae is used to receive signals.

During the signal acquisition process, we first collect 1000 samples at each reference point. We then construct an imbalanced fingerprint database according to different imbalance ratios. Here, the imbalance ratio refers to the proportion of the number of samples in the minority class to the number of samples in the majority class. We set three imbalance ratios in our experiments, which are 1:5, 1:10, and 1:20, respectively. Namely, the number of samples in each minority class is 200, 100, and 50, respectively. In order to verify the impact of unbalanced distribution of fingerprint data on localization accuracy, we use two deep learning models 1D-MobileNet and DeepFi [32] to locate fingerprint data signal at different imbalance rates. Among them, DeepFi is a representative method for indoor localization of fingerprint data. Furthermore, 1D-MobileNet is our own improved method based on the lightweight model MobileNet v2 [33], and its principle and architecture will be discussed in Section 4.3.

The experimental results are shown in Figure 2. It can be seen that with the increase in the imbalance rate of the fingerprint database, the location accuracy of the two methods decreases continuously. This shows that the unbalanced distribution of fingerprint data has a great impact on the later location accuracy. This further verifies that the background and motivation of our proposed research topic are reasonable. Therefore, how to effectively solve the imbalance problem of the fingerprint database is the key research issue.

### 3.2. Preliminaries on Channel State Information

Thanks to the Inter Wireless Link 5300 NICs, channel state measurements are now easier to perform than in the past. CSI can now be collected from a computer by installing the specified driver. The CSI reflects the channel changes that occur during transmission. WiFi signals propagated in complex environments can suffer significant losses due to multipath effects, fading, shadowing, and delay distortion. CSI is critical for presenting channel properties in real-world scenarios.

In a channel with narrowband flat fading, the channel information is modeled in the frequency domain as follows:(1)$$\vec{y}=H\vec{x}+\vec{n},$$
where: $\vec{x}$ and $\vec{y}$ denote the frequency domain transmit data vector and the frequency domain receive data vector, respectively, H is the channel state information is the complex matrix, and $\vec{n}$ is the additive Gaussian white noise vector.

For a WLAN with a MIMO system, each packet has a complex matrix of *p* × *q* × 30 $H_{MIMO}$, where *p* is the number of transmit antennas and *q* is the number of receive antennas. *m* = *p* × *q* is the number of antenna pairs.
(2)$$H_{MIMO}=\left[\begin{array}{cccc} H_{11} & H_{12} & \cdots & H_{1q}\\ H_{21} & H_{22} & \cdots & H_{2q}\\ \vdots & \vdots & \vdots & \vdots \\ H_{p1} & H_{p2} & \cdots & H_{pq} \end{array}\right],$$

The CSI for the 30 sub-channels is as follows.
(3)$$H=\left[h_{1},h_{2},h_{3},\cdots ,h_{30}\right],$$
(4)$$h_{i}=\left|h_{i}\right|e^{j\sin\left(∠h_{i}\right)},$$

In our experiment, the transmitter is one antenna and the receiver is three antennas, so the dimension of each packet is 1 × 3 × 30, as shown in Figure 3.

### 3.3. Workflow of the Csi Fingerprint Localization System

In general, the overall workflow of the imbalanced CSI fingerprint-based localization system can be divided into an offline stage and online stage (see Figure 4). In this subsection, we separately describe the offline and online stages in detail to understand the idea of the whole work.

#### 3.3.1. Offline Stage

For the offline stage, it is necessary to select reference points (RPs) in the location area, and the RPs are required to be distributed as evenly as possible and cover the entire area to be located. We first collect fingerprint data at different RPs and preprocess the data to extract different numbers of samples. In this way, we build an imbalanced fingerprint database, which is required by the topic of our study. Then, our proposed oversampling method is used to specifically deal with this imbalanced database. After being processed by the oversampling method, we obtain a relatively complete standard fingerprint database, in which the quantity of each class of data tends to be consistent. Finally, the processed fingerprint database is fed to the model for training, and a location model with good performance is obtained. Here, we use two models, DeepFi and an improved 1D-MobileNet, as localization models, respectively. Next, the trained positioning model is tested in the online stage to further verify the performance of our proposed oversampling method for solving imbalanced database problems.

#### 3.3.2. Online Stage

The verification process is mostly performed in the online phase, when a randomly selected RPs point measures its CSI data and enters it into an offline fingerprint database for identification, which can evaluate the present location and validate the positioning system’s correctness. The offline stage is to verify the accuracy of the positioning system. In the test stage, all reference points are regarded as access points, and the CSI data are measured and collected at each access point as test data. It is then fed into an offline fingerprint database for matching and identification to assess its current location, thus verifying the accuracy of the positioning system.

Throughout the workflow, it can be seen that the oversampling method can be considered as a preprocessing method before the localization model. Most of the existing localization methods currently use deep learning to train offline fingerprint databases but ignore the importance of data class balance in fingerprint databases. Therefore, it is crucial to adopt a suitable oversampling method that can reduce the skew of data classes while capturing their main features. We will describe the proposed oversampling method in detail in the Section 4.

## 4. Our SASMOTE Model and Training Scheme

Based on the SMOTE algorithm, we put forward a self-attention encoder-decoder framework to make it easier to supplement data of the fingerprint database and reduce complex data collection efforts. In this section, we first introduce the principles of the SMOTE algorithm and the proposed SASMOTE framework. Then, different evaluation criteria are described to assess the performance of the proposed approach.

### 4.1. Smote Algorithm

The algorithm, Synthetic Minority Oversampling Technique (SMOTE) [18], is a typical oversampling method to solve the problem of data imbalance. Its primary idea is to construct new minority instances by combining many existing minority instances. Specifically, its working principle is as follows: (1) for each sample $x_{i}$ of the minority class, calculate its Euclidean distance to all samples in the minority class sample set *S*, and obtain its *k* nearest neighbors (KNNs). (2) The sampling ratio *n* is determined according to the imbalanced proportion of samples. For each minority class sample $x_{i}$, one sample $x_{old}$ is randomly selected from *k* nearest neighbor samples of the same class as the auxiliary sample for the synthesis of new samples. (3) For each randomly selected neighbor $x_{old}$, a new sample $x_{new}$ is generated with $x_{i}$ according to the following random interpolation formula, and an interpolation sample is finally synthesized.
(5)$$x_{new}=x_{i}+\mathrm{rand}\left(0,1\right)\left(x_{i}-x_{old}\right),$$
where rand (0, 1) represents a random value in the (0, 1) interval. By repeating this process multiple times, multiple samples can be generated to balance the dataset. Figure 5 shows a schematic diagram of using the SMOTE algorithm to generate new samples.

SMOTE algorithm is a classical oversampling algorithm which synthesizes a few classes. It can effectively solve the overfitting problem, which easily occurs in traditional sampling methods. In this way, it can effectively solve the overfitting problem caused by random oversampling. Therefore, we consider integrating the SMOTE algorithm in our proposed framework to solve the imbalance problem of the fingerprint database.

### 4.2. SASMOTE Model

However, it is found in experiments that the traditional SMTOE algorithm is suitable for samples with low-dimensional and simple features, but it is not suitable for processing high-dimensional data, such as CSI fingerprint data. Therefore, we propose a novel SAMOTE framework (see Figure 6) to solve this problem. Specifically, we first utilize a self-attention encoder to extract fingerprint data features and compress the dimensionality of fingerprint data. Then, we utilize the SMOTE algorithm to generate fresh data and output them through the decoder.

#### 4.2.1. Self-Attention Encoder-Decoder

In the self-attention encoder-decoder, the adopted attention module is inspired by SAGAN, which is proposed by Zhang et al. [34]. Multiple convolutions are required to generate global dependencies because the model has just convolution and most of the convolution kernels are 1 × 1 and 3 × 3, and the receptive field is too tiny. The use of a self-attention method to obtain dependencies at one layer at a distance rather than a multi-layer convolution operation is particularly successful for one-dimensional long sequence data and it minimizes processing effort. As shown in Figure 7, transposing f(x) and multiplying it with g(x), as well as providing parameters to multiply with the attention map and superimposing it with the input *x*, yields the attention map. Finally, the output is as follows:(6)$$y_{i}=\gamma o_{i}+x_{i},$$
where γ is the scale parameter and is initialized to 0. The network learns the local information first and then learns the remote information slowly, i.e., from easy to hard. We integrate this attention mechanism module into each layer of the encoder and decoder, respectively, so that the network has stronger feature representation ability.

#### 4.2.2. Enhanced Loss Function

The loss function is used in this study to calculate the degree of inconsistency between the data and the original data after encoding and decoding, and it is a crucial component of codec training. SMOTE is based on the KNN regression model’s premise and examines whether to use the L_1_ or L_2_ loss function for evaluation. The distinction between L_2_ and L_1_ loss is that L_2_ loss squares the distance between the estimated and true values, imposing a substantial penalty on the output that differs from the observed values. The absolute value of the difference between the estimated and real values is used in the L_1_ loss, which is insensitive to output that differs from the genuine value. So, it is beneficial to keep the model stable when there are outliers in the observation. The L_1_ loss function was chosen because there are several outliers in the CSI measurement.
(7)$$\begin{array}{cc} L_{2}\;\mathrm{loss}\;:\; & L\left(y,\hat{y}\right)=1/n\sum {\left(\hat{y}-y\right)}^{2},\\ L_{1}\;\mathrm{loss}\;:\; & L\left(y,\hat{y}\right)=1/n\sum \left|\hat{y}-y\right|, \end{array}$$
where $\hat{y}$ represents the predicted value and *y* represents the true value. A batch contains *n* elements.

### 4.3. Evaluation Model and Metrics

#### 4.3.1. Location Estimation Model

The standard classification model has a big memory need and a large number of operations; thus, it cannot be employed as a localization model for indoor mobile needs. Therefore, we also propose an improved 1D-MobileNet for the identification and localization of fingerprint data. The improvement of the 1D-MonileNet model is based on the MobileNet V2 model, and the biggest difference is that the input and output data of our model are one-dimensional data. Figure 8 depicts the basic structure of our 1D-MobileNet model. At the input and output ends of the framework, we first use 1 × 1 convolution to transform the high-dimensional CSI fingerprint data into one-dimensional data. It is then fed into an intermediate feature extraction module, which is stacked with seven MobileNet V2 modules.

#### 4.3.2. Evaluation Metrics

The most essential measurement index for a localization system is the system’s localization accuracy. The localization system’s performance is measured using the following indicators.

As a measure of the accuracy of the localization algorithm, we use the Cumulated Density Function (CDF). A significant number of localization tests are used to assess the system’s stability from the standpoint of the probability distribution of the localization error. The stability of the system is defined in Equation (Equation 8), where P(X≤x) represents the probability that the localization error is less than *x*
(8)$$F_{x}\left(x\right)=P\left(X\leq x\right).$$

The maximum localization error is used as a measure of the stability and robustness of the positioning system, which is defined as the the maximum error of the positioning system after all tests, and its expression is,
(9)$$\;\mathrm{MaxError}\;=\arg\max\sqrt{{∥p_{i}\left(t\right)-{\hat{p}}_{i}\left(t\right)∥}^{2}}.$$

Average Root Mean Square Error (ARMSE) was utilized as a parameter measure to evaluate the localization algorithm’s accuracy. This index measures the localization accuracy of the localization system from the perspective of evaluating the localization error, which is defined as,
(10)$$\mathrm{ARMSE}=\frac{1}{N}\sum_{i=1}^{N}\sqrt{{\left|p_{i}\left(t\right)-{\hat{p}}_{i}\left(t\right)\right|}^{2}},$$
where $p_{i}\left(t\right)$ denotes the actual two-dimensional coordinates of the *i*-th test position and ${\hat{p}}_{i}\left(t\right)$ are the *i*-th test position’s location estimate 2D coordinates; *N* is the total number of position tests.

## 5. Experiments

The proposed oversampling method’s performance is examined in this section. SASMOTE’s performance is compared to those of other oversampling approaches. The fresh fingerprint database created by the SASMOTE model is then fed into other classifiers to acquire separate findings in order to assess the SASMOTE model’s applicability.

### 5.1. Experimental Setup

As illustrated in Figure 1, our experiments are performed in a classroom. There are 25 cumulative RPs with an average interval of 0.6 m in a 10 m × 6 m classroom. Because of the walls, desks, and shelves, the experimental setting is complicated, and there are several Non-Light Of Sight (NLOS) RPs. To avoid CSI outliers acquired due to the complicated environment impacting the experimental results, we used the middle RPs as the minority class points of the unbalanced fingerprint database. We set up a wireless router with an antenna as the transmitter in IEEE 802.11n AP mode in the classroom, as well as a laptop computer with an Intel Wireless Link 5300 NIC. On the laptop, we installed Ubuntu 16.04 and changed the WiFi driver kernel. The actual CSI data may now be transported to the laptop and read with the Linux 802.11n CSI tool in the latest kernel.

In this experiment, we collect approximately 1500 CSI samples at each RP, and the dimension of each CSI packet is 1 × 3 × 30. To construct an unbalanced fingerprint database, 1000 sample data are chosen as training samples for the majority class points, and 200, 100, and 50 sample data are chosen for the minority class points. In addition, 500 samples of each class are used as the test set to verify the method performance. Then, we use the created imbalanced fingerprint database to train the self-attention encoder-decoder of the SASMOTE and supplement the fingerprint database with the SASMOTE model, as shown Algorithm 1.
**Algorithm 1** SASMOTE model.**Require:** Batches of imbalanced CSI fingerprints: $I=\left\{i_{1},i_{2},\ldots ,i_{n}\right\}$; number of minority classes: *k*; Model parameters: $\theta =\left\{\theta_{1},\theta_{2},\ldots ,\theta_{j}\right\}$; Learning Rate: α;**Ensure:** Balanced CSI data of the minority class point: *S*; **Train the Encoder/Decoder:**1: **for** epoch ∈[1,maxepoch] **do**2:   $E_{I}\leftarrow encode\left(I\right)$;3:   $D_{I}\leftarrow encode\left(E_{I}\right)$;4:   $R_{L}=\frac{1}{n}\sum_{i=1}^{n}\left|D_{Ii}-I_{i}\right|$;5:   $P_{I}\leftarrow permute\_order\left(I\right)$;6:   $E_{P}\leftarrow encode\left(P_{I}\right)$;7:   $D_{P}\leftarrow decode\left(E_{P}\right)$;8:   $B_{L}=\frac{1}{n}\sum_{i=1}^{n}\left|D_{Pi}-P_{Ii}\right|$;9:   $T_{L}=R_{L}+B_{L}$;10:  $\theta =\theta -\alpha \frac{\partial T_{L}}{\partial \theta }$;11: **end for** **Generate Simple:**12: **for** j∈[1,k] **do**13:  $C\leftarrow select(B_{j})$;// Select minority classes from the fingerprint database;14:  E←encode(C);15:  G←SMOTE(E);16:  S←decode(G);17: **end for**

Figure 9 shows the process of training the self-attention encoder-decoder of SASMOTE. When the 90-dimensional CSI data are fed into the SASMOTE encoder, it is compressed into six-dimensional data by four convolutional layers fused with the self-attention module. Instead, the SASMOTE decoder is composed of ConvTranspose layers (i.e., deconvolution layer [35]) and self-attention layers to extend the data of dimension 6 to 90. Furthermore, the L_1_ loss function is used to calculate the variances of the obtained data. The imbalanced fingerprint database is disrupted again, and the same action is repeated. The model’s learning rate was set to 0.0002, the training batch size to 100, and the total number of epochs to 200.

### 5.2. Localization Performance

First, we use 1D-MobileNet for localization experiments, as shown in Figure 10, which shows the CDF curves of localization errors for fingerprint databases with different imbalance ratios. The red line shows the localization accuracy of the fingerprint database with imbalance ratio 5 and also the CDF curve with the best result. Its ARMSE distance is about 0.85 m, the minimum error distance is about 0.06 m, and the MaxError distance is about 2.64 m. Its positioning error distance is 64% around 1 m, 87% around 2 m, and 99% within 3 m. When the unbalance ratio is 20, the positioning error is obviously the largest. In this case, its ARMSE distance is about 1.34 m, the minimum error distance is about 0.09 m, and the MaxError distance is 3 m. In addition, its positioning error distance is 33% around 1 m, 69% around 2 m, and 99% within 3 m. We discovered that when the imbalance ratio is too high, the generally utilized localization algorithms overfit, and the loss rises throughout training, as shown in Figure 11. Compared with the DeepFi localization method, the 1D-MobileNet training loss is more stable.

Next, we train the self-attention encoder-decoder of the SAMOTE. For comparison, we created a class distribution of the original fingerprint database and a class distribution after the SAMOTE encoder, as shown in Figure 12. The 2D projections of the unprocessed CSI data are clearly mixed together, with the features obscured. The data following the SA encoder has more noticeable features, demonstrating the effectiveness of using the SA encoder before the SMOTE oversampling approach. We were also able to extract the confusion matrix of the unbalanced fingerprint database positioning and the balanced fingerprint database positioning. As can be seen from Figure 13, in the initial fingerprint database identification process, the misjudgment is more serious, and the predicted value will be biased towards the majority of the points due to imbalance, as shown in the fifth row in Figure 13a. In contrast, the balanced fingerprint library recognition is relatively stable.

Finally, we use SASMOTE to complement the imbalance fingerprint database and compare it with the existing oversampling approaches, e.g., Gaussian_SMOTE [16] and Kmeans_SMOTE [17]. In addition, several good oversampling methods that have appeared recently are also used for comparison, such as NANSMOTE [23], FCSMI [24], and ASNSMOTE [25]. As shown in Figure 14, processed by various oversampling approaches, the CDF curves of fingerprint databases with different imbalance ratios are indicated, with the black line being the initial imbalance fingerprint database CDF curve. As shown in Figure 14a, the CDF curve of the fingerprint database with an imbalance ratio of 5 is shown after the oversampling process. The red curve represents the CDF curve using the Gaussian_SMOTE method. The probability that the error distance is within 1 m is 70%, the probability of being within 1.5 m is 89%, and the probability of not exceeding 2 m is 93%. The blue curve represents the CDF curve using the Kmeans_SMOTE method, the probability of error distance is less than 1 m is 67%, the probability of less than 1.5 m is 71%, and the probability of not more than 2 m is 79%. The green curve represents the CDF curve using the NANSMOTE method, the probability of error distance is less than 1 m is 64%, the probability of being less than 1.5 m is 81%, and the probability of not exceeding 2 m is 91%. The cyan curve represents the CDF curve using the FCSMI method. The probability of the error distance being less than 1 m is 66%, the probability of it being less than 1.5 m is 92%, and the probability of it not exceeding 2 m is 96%. The yellow curve represents the CDF curve using the ASNSMOTE method. The probability of the error distance being less than 1 m is 65%, the probability of it being less than 1.5 m is 90%, and the probability of it not exceeding 2 m is 94%. The purple curve represents the CDF curve using the SASMOTE method, the probability of error distance being less than 1 m is 64%, the probability it is less than 1.5 m is 94%, and the probability it is not more than 2 m is 98%. It can be seen that when the imbalance ratio is 5, the errors of each oversampling method are not much different. The worst performance is Kmeans_SMOTE, even when the error distance is greater than 1.2 m; the effect is not as good as the initial fingerprint library. The best performer is our proposed SASMOTE.

As shown in Figure 14b, the CDF curve of the fingerprint database with an imbalance ratio of 10 is shown after the oversampling process. Using the CDF curve of the Gaussian_SMOTE method, the probability of error distance within 1 m is 59%, the probability within 1.5 m is 83%, and the probability of not exceeding 2 m is 91%. Using the CDF curve of the Kmeans_SMOTE method, the probability of the error distance being less than 1 m is 67%, the probability of it being less than 1.5 m is 77%, and the probability is it not more than 2 m is 86%. Using the CDF curve of the NANSMOTE method, the probability of the error distance being less than 1 m is 57%, the probability of it being less than 1.5 m is 79%, and the probability of it not exceeding 2 m is 88%. Using the CDF curve of the FCSMI method, the probability of the error distance being less than 1 m is 60%, the probability of it being less than 1.5 m is 84%, and the probability of it not exceeding 2 m is 93%. Using the CDF curve of the ASNSMOTE method, the probability of the error distance being less than 1 m is 60%, the probability of it being less than 1.5 m is 84%, and the probability of it not exceeding 2 m is 92%. Using the CDF curve of the SASMOTE method, the probability of the error distance being less than 1 m is 62%, the probability of it being less than 1.5 m is 89%, and the probability of it not exceeding 2 m is 96%. It can be seen that, affected by the unbalanced ratio, the overall positioning accuracy decreases. Generally, when the error distance is about 2.8, the CDF reaches 1.

As shown in Figure 14c, the CDF curve of the fingerprint database with an imbalance ratio of 20 is shown after the oversampling process. Using the CDF curve of the Gaussian_SMOTE method, the probability of error distance within 1 m is 38%, its probability within 1.5 m is 68%, and its probability of not exceeding 2 m is 74%. Using the CDF curve of the Kmeans_SMOTE method, the probability of the error distance being less than 1 m is 56%, the probability of it being less than 1.5 m is 62%, and the probability of it not exceeding 2 m is 69%. Using the CDF curve of the NANSMOTE method, the probability of the error distance being less than 1 m is 42%, the probability of it being less than 1.5 m is 54%, and the probability of it not exceeding 2 m is 71%. Using the CDF curve of the FCSMI method, the probability of the error distance being less than 1 m is 48%, the probability of it being less than 1.5 m is 72%, and the probability of it not exceeding 2 m is 81%. Using the CDF curve of the ASNSMOTE method, the probability of the error distance being less than 1 m is 43%, the probability of it being less than 1.5 m is 66%, and the probability of it not exceeding 2 m is 74%. Using the CDF curve of the SASMOTE method, the probability of the error distance being less than 1 m is 60%, the probability of it being less than 1.5 m is 88%, and the probability of it not exceeding 2 m is 94%. It can be seen that when the imbalance ratio is 20 and the number of samples of the minority class is 50, the traditional SMOTE method is greatly affected, and the positioning accuracy is greatly reduced. Because our proposed SASMOTE uses SA encoder-decoder training before generating data, the imbalance ratio of the initial fingerprint library has little effect on SASMOTE, as shown in Figure 15.

Table 1 shows the localization error results obtained by using 1D-MobileNet with different oversampling methods under different fingerprint imbalance ratios. This is mainly through ARMSE, minimum error, and maximum error, evaluating their positioning accuracy. The smaller the error, the better the positioning effect.

In addition, we also evaluate the new fingerprint database generated by SASMOTE with different methods, which contain our method (1D-MobileNet), DeepFi, and FIFS [36], as shown in Table 2. After complementing the fingerprint database with an unbalanced ratio of 5, the ARMSE distance is 0.13 m using the FIFS localization method. In this case, the minimum error distance is 0.02 m and the MaxError distance is 2.84 m. Using the DeepFi localization method, the ARMSE distance is 0.91 m. In this case, the minimum error distance is 0.02 m and the MaxError distance is 2.74 m. After complementing the fingerprint database with an unbalanced ratio of 10, the ARMSE distance is 1.17 m using the FIFS localization method. In this case, the minimum error distance is 0.02 m and the MaxError distance is 2.91 m. Using the DeepFi localization method, the ARMSE distance is 0.96 m. In this case, the minimum error distance is 0.02 m and the MaxError distance is 2.78 m. After completing the fingerprint database with an unbalanced ratio of 20, the ARMSE distance is 1.21 m using the FIFS localization method. In this case, the minimum error distance is 0.02 m and the MaxError distance is 3 m. Using the DeepFi localization method, the ARMSE distance is 1.03 m. In this case, the minimum error distance is 0.02 m and the MaxError distance is 2.83 m. So, employing the novel oversampling method, SASMOTE is still applicable in the presence of other localization methods.

## 6. Conclusions

In this article, we proposed a novel deep learning-based oversampling method, the Self-Attention Synthetic Minority Oversampling Technique (SASMOTE), called SAMOTE, to solve the imbalanced fingerprint database in WiFi localization. SASMOTE fuses the highly popular self-attention module from [34] and SMOTE algorithm [18]. We design a multi-layer encoder-decoder to compress and recover high-dimensional CSI fingerprint data features. Among them, each layer of encoder and decoder is integrated with a self-attention module to help the model enhance its feature representation ability. Then, the SMOTE algorithm is used to synthesize the minority class samples to make the database achieve class balance. Additionally, we specially construct an imbalanced fingerprint database for training and validation of the proposed model. In the verification stage, we also propose an improved localization model, 1D-MobileNet, to verify the localization accuracy of fingerprints on the processed balanced database, thereby further evaluating the superiority of our algorithm.

Numerous experimental studies have shown that SASMOTE outperforms many existing oversampling methods. In the case of imbalanced fingerprint database caused by removing outliers or missing fingerprints, the method can effectively generate CSI data with similar features to minority class RP to complement its fingerprint database. Furthermore, the performance of fingerprint databases generated by SASMOTE is stable when tested in different localization method models. We believe that the key to the success of SASMOTE lies in the efficient feature extraction of high-dimensional fingerprint data by the self-attention encoder-decoder. This enables the classical oversampling method to generate higher quality samples to complement the imbalanced database. Unlike GAN, which is biased to generate 2D feature images, our proposed method attempts an end-to-end implementation of CSI data. That is, the original features of the data are retained, which is more conducive to the application of the localization system in mobile.

The purpose of our article is to propose an idea based on deep learning to effectively solve the problem of fingerprint data imbalance encountered in practice. However, this research topic has much room for improvement in the future. First, our experimental environment is limited, and the proposed algorithm cannot guarantee stable performance in different experimental environments. In addition, with the development of deep learning technology, it is a meaningful direction to use better models to directly solve the imbalance problem of the fingerprint database in an end-to-end manner. In the future, we will develop our work around these two points.

## Figures and Tables

**Figure 1 sensors-22-05677-f001:**
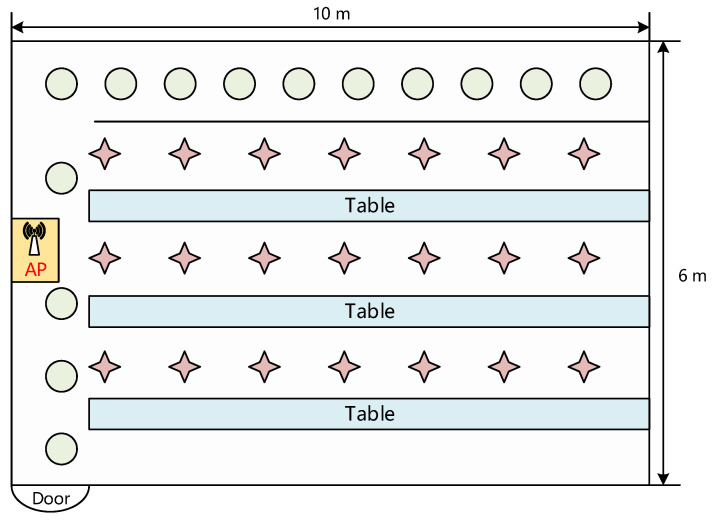
Illustration of the experimental environment (a classroom). The circles in the figure denote the reference point of the majority class fingerprints data, and the stars denotes the reference point of the minority class fingerprints data.

**Figure 2 sensors-22-05677-f002:**
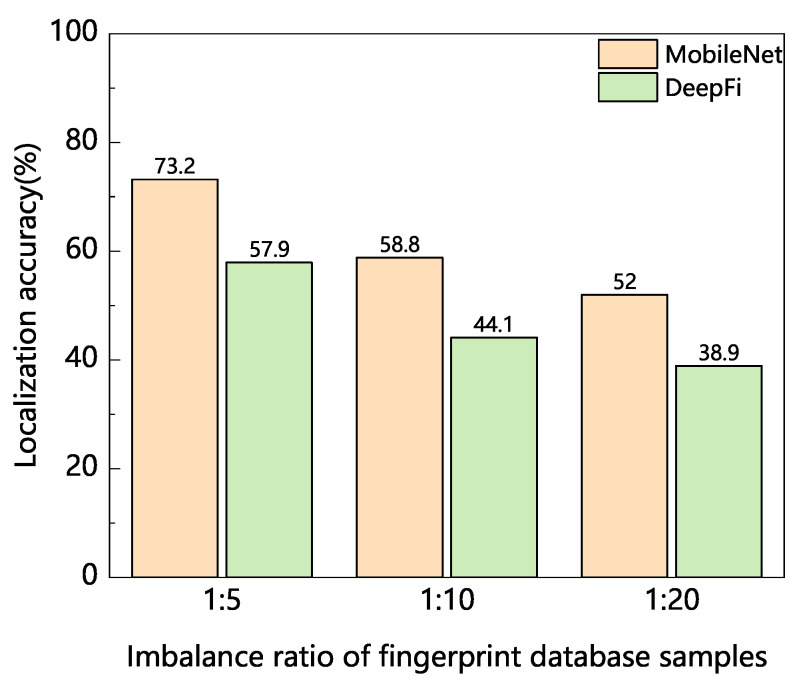
The influence of different imbalance rates of fingerprint database on localization accuracy.

**Figure 3 sensors-22-05677-f003:**
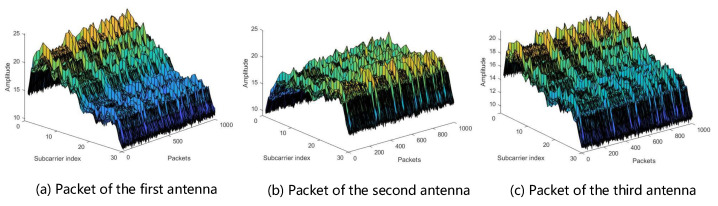
Packets with different antennas. *x*-axis is the number of packets, *y*-axis is the subcarrier length, and *z*-axis is the signal amplitude.

**Figure 4 sensors-22-05677-f004:**
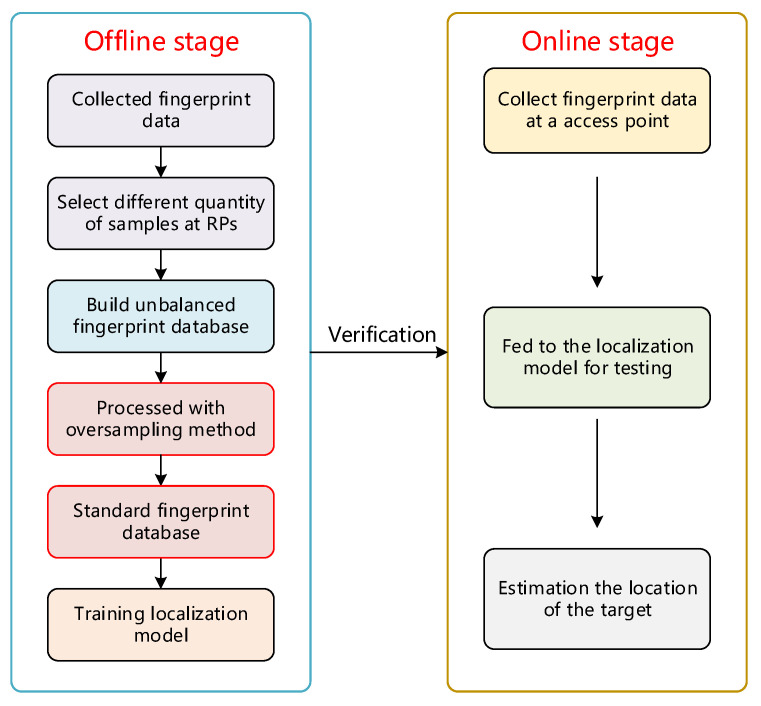
Structure of CSI fingerprint positioning system.

**Figure 5 sensors-22-05677-f005:**
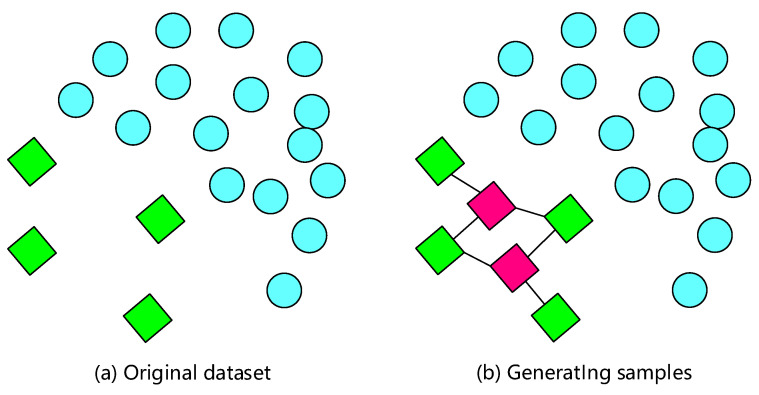
Schematic diagram of new sample generation using SMOTE algorithm. Blue circles represent the majority data, green squares represent the minority data, and red squares represent generated data.

**Figure 6 sensors-22-05677-f006:**
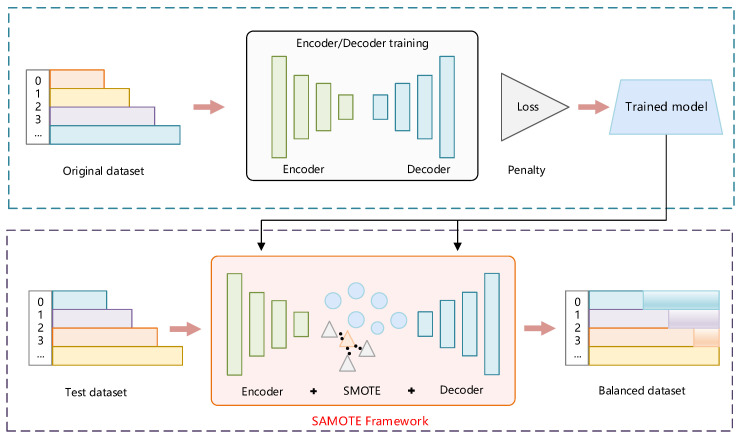
Illustration of SASMOTE implementation. In the first part, training the SA encoder and decoder; in the second part, using the trained encoder/decoder to complete the fingerprints.

**Figure 7 sensors-22-05677-f007:**
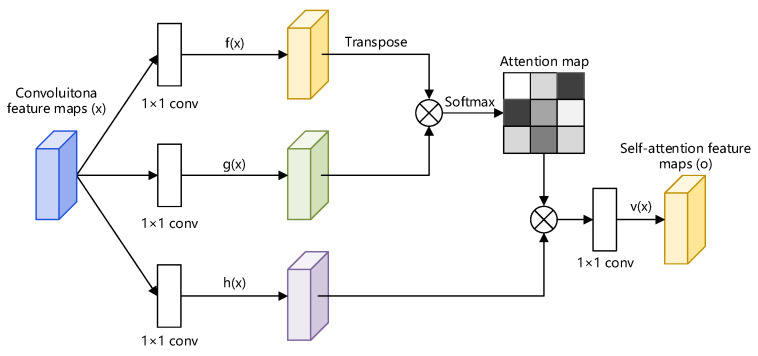
The proposed self-attention module for the Encoder/Decoder. The ⊗ denotes matrix multiplication. The softmax operation is perfo.

**Figure 8 sensors-22-05677-f008:**
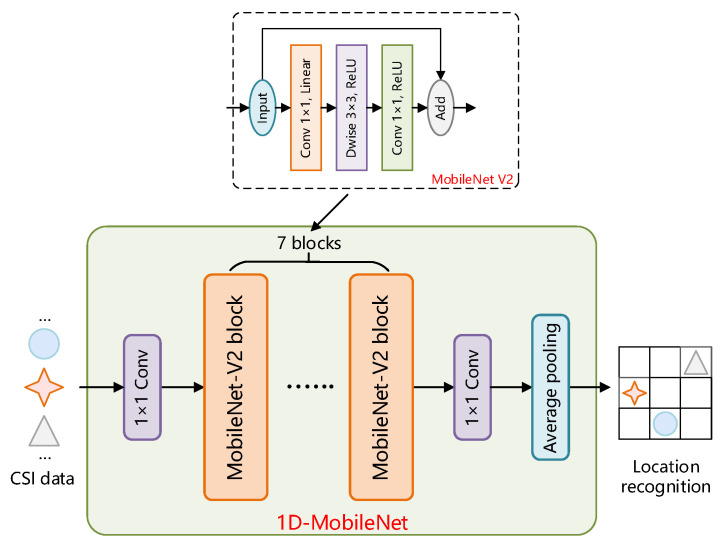
The structure of the base block of the model; the structure contains shortcut connections. In addition, satisfying stride = 1 and inputting feature matrix and output feature matrix of the same shape will skip the base structure block to obtain the output. The different symbols in the figure represent fingerprint maps.

**Figure 9 sensors-22-05677-f009:**
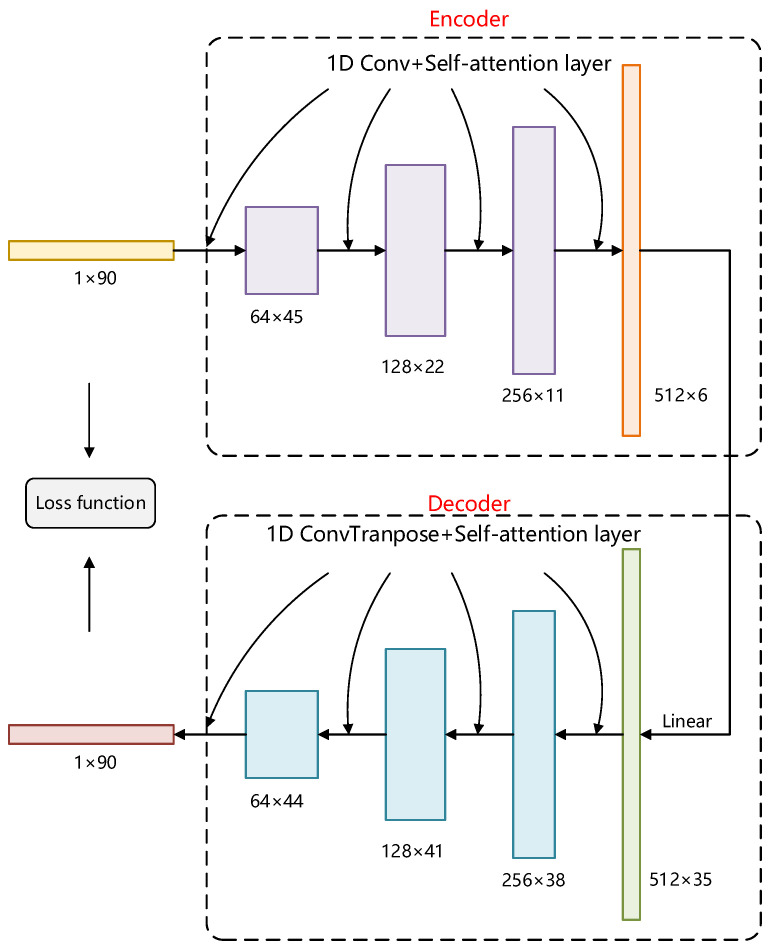
Schematic of the training of the self-attention encoder-decoder. The encoder is used to extract fingerprint data features, and the decoder is used to restore the extracted fingerprint data.

**Figure 10 sensors-22-05677-f010:**
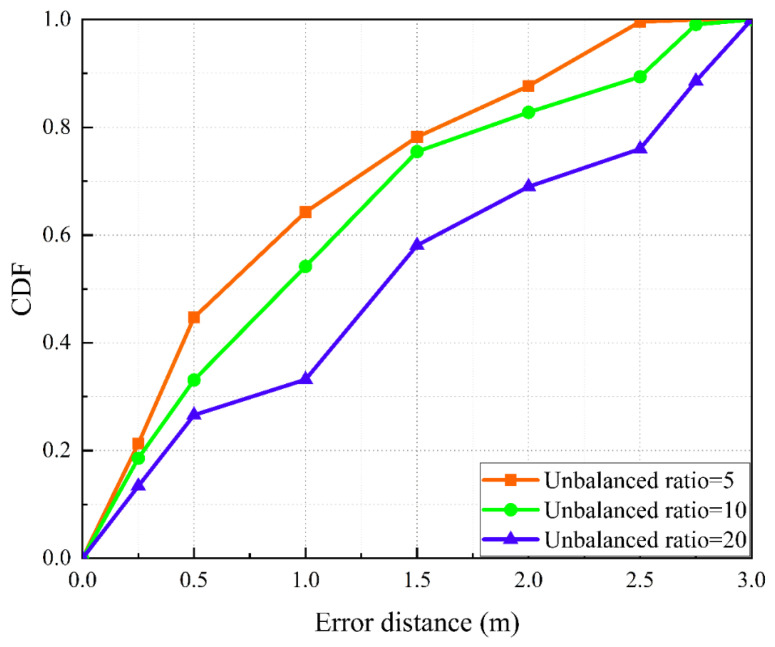
Initial database localization results. The red line indicates a fingerprint database with an imbalance ratio of 5, the green line indicates a fingerprint database with an imbalance ratio of 10, and the blue line is a fingerprint database with an imbalance ratio of 20.

**Figure 11 sensors-22-05677-f011:**
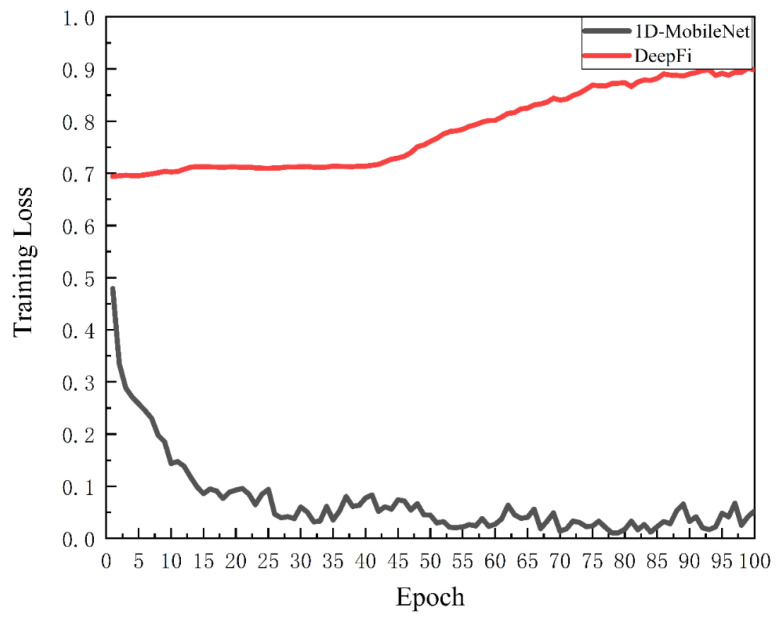
Training loss results for different localization methods.

**Figure 12 sensors-22-05677-f012:**
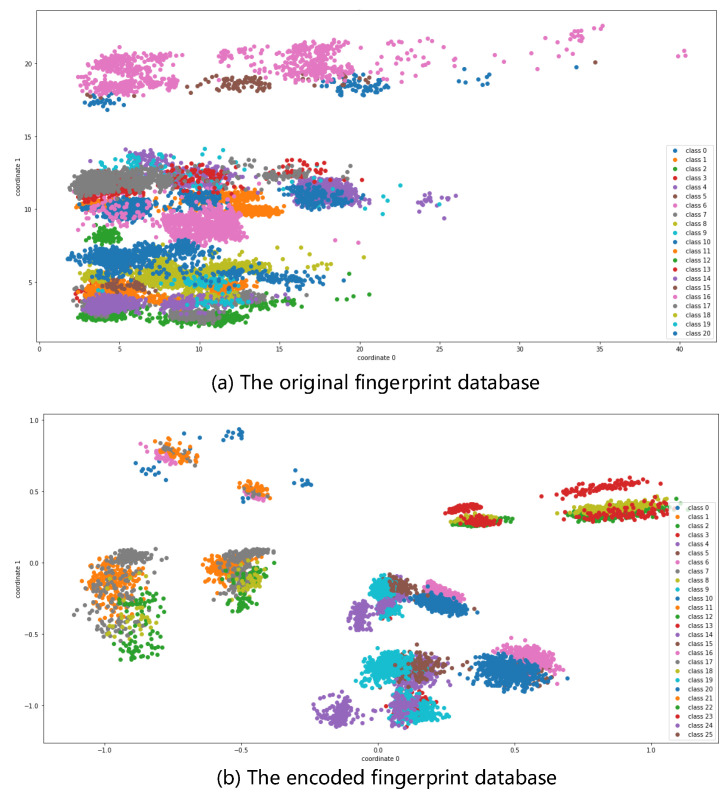
2D projection of the initial fingerprint database and the encoded fingerprint database.

**Figure 13 sensors-22-05677-f013:**
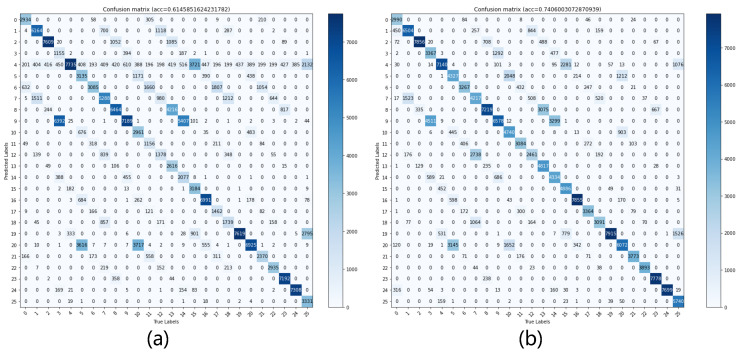
Confusion matrix of localization recognition results for the original database and balanced database. Subfigures (**a**) shows the location identification result with unbalanced fingerprint, and subfigures (**b**) shows the identification result after SASMOTE processing.

**Figure 14 sensors-22-05677-f014:**
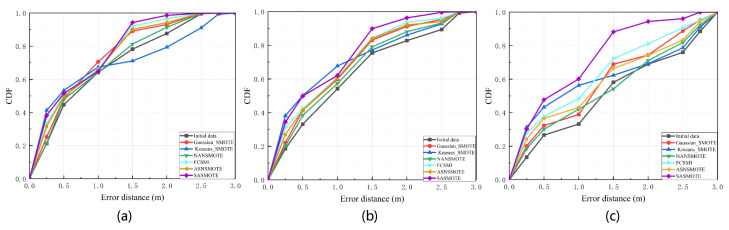
Fingerprint database after different oversampling methods. (**a**) The localization results of fingerprint database with an imbalanced ratio of 5. (**b**) The localization results of fingerprint database with an imbalanced ratio of 10. (**c**) The localization results of fingerprint database with an imbalanced ratio of 20.

**Figure 15 sensors-22-05677-f015:**
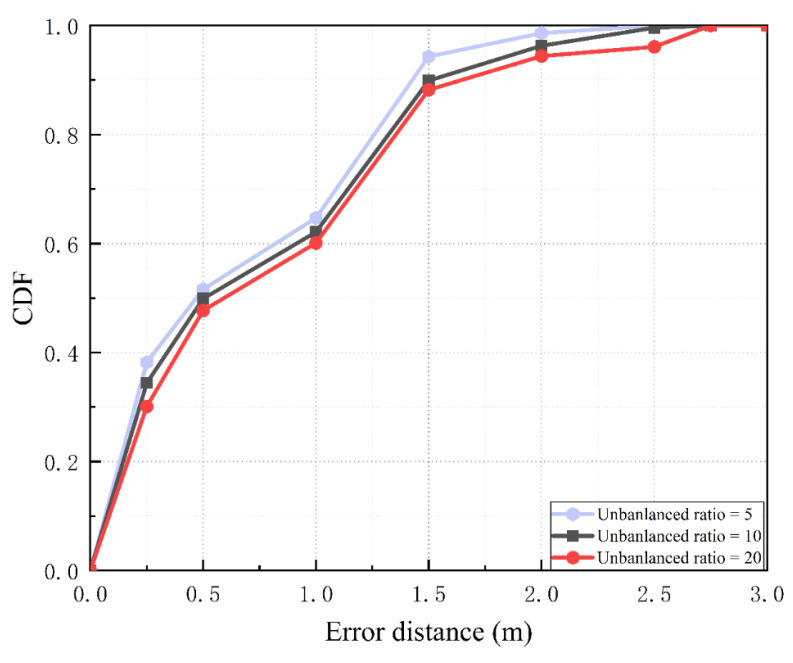
The localization results of the fingerprint database after SASMOTE.

**Table 1 sensors-22-05677-t001:** Location error of different oversampling methods.

Method	Unbalanced Ratio:5			Unbalanced Ratio:10			Unbalanced Ratio:20		
	ARMSE	Min.	Max.	ARMSE	Min.	Max.	ARMSE	Min.	Max.
Initial dataset	0.85	0.06	2.64	1.14	0.07	3.00	1.54	0.09	3.00
Gaussian_SMOTE	0.74	0.06	2.51	0.84	0.06	2.75	1.28	0.06	2.97
Kmeans_SMOTE	0.91	0.03	2.68	0.95	0.03	2.84	1.34	0.03	3.00
NANSMOTE	0.79	0.04	2.54	0.86	0.04	2.78	1.40	0.04	3.00
FCSMI	0.72	0.03	2.43	0.80	0.03	2.67	1.18	0.03	3.00
ASNSMOTE	0.76	0.04	2.64	0.89	0.04	2.79	1.27	0.04	3.00
SASMOTE	0.70	0.02	2.53	0.72	0.02	2.57	0.76	0.02	2.62

**Table 2 sensors-22-05677-t002:** Location error of different models after SASMOTE completed the fingerprints.

Method	Unbalanced Ratio:5			Unbalanced Ratio:10			Unbalanced Ratio:20		
	ARMSE	Min.	Max.	ARMSE	Min.	Max.	ARMSE	Min.	Max.
FIFS	1.13	0.02	2.84	1.17	0.02	2.91	1.21	0.02	3.00
DeepFi	0.91	0.02	2.74	0.96	0.02	2.78	1.03	0.02	2.83
1D-MobileNet	0.70	0.02	2.53	0.72	0.02	2.57	0.76	0.02	2.62

## Data Availability

Not applicable.

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
