# Peer review of "SASMOTE: A Self-Attention Oversampling Method for Imbalanced CSI Fingerprints in Indoor Positioning Systems"

_sensors, 2022, doi:10.3390/s22155677_

Round 1

Reviewer 1 Report

1.     Future works should be mentioned at the end of conclusion

2.     In the discussion part,  in order to support the new idea of this paper, the relative comparison or discussion should be added on the technology between this paper and the above given references.

3.     Contributor needs to justify, how the proposed system is increasing the intelligence.

4.     Narrate the common challenges can be included in the introduction section?

5.     The   aim, the motivation and the contribution of the paper should be emphasized at the end of introduction

6.     Updating the related work section needs to highlight the challenges and limitations of the regular mechanism.

7.     In this article, the abstract is too narrow, try to give some clarification about the specification

8.     In this overall article, contributors made several punctuation errors. So, it must be avoided.

9.     Experimental result part is too weak, need more  with brief explanation with comparison analysis, what you have achieved using the proposed one

10.The betterment of the proposed mechanism is well stated. Still, the paper needs to state the effectiveness of the proposed mechanism.

11.Include merits and demerits of existing research works, as it helps readers understand the techniques which is used .

12.The primary objectives are defined in related work, but the author failed to provide the achieved performance in terms of statistical values.

13.Proposed method needs more attention

14. Explanation needed for used formulas/equations

15. Elaboration of the SASMOTE-based strategy is in need of  more justification.

16.I recommend the author to highlight key contributions of the paper

17.Figure 9 and 10 are not up-to the level, brief details can be provided

18.Literature review needs more attention,some papers only cited in literature review.

Reviewer 2 Report

1. Related works section may include literature survey.

2. Cite the reference sequentially.

3. Add/cite recent publication (2019, 2020, 2021) preferably.

4. A comparative study may also be shown in graphical form.

5. Abstract should be precise and should clearly show the background, hypothesis, problem statements, methodology, techniques used with relevant method, and the results obtained.

Reviewer 3 Report

1. The references are out-of-date. Authors should add more recent studies and compare the proposed scheme with the existing schemes. The author should consider the following recent and closely related papers

A. J. Hoyos-Osorio, A. Alvarez-Meza, G. Daza-Santacoloma, A. Orozco-Gutierrez, G. Castellanos-Dominguez, Relevant information undersampling to support imbalanced data classification, Neurocomputing,Volume 436,2021,Pages 136-146

B. Junnan Li, Qingsheng Zhu, Quanwang Wu, Zhu Fan,A novel oversampling technique for class-imbalanced learning based on SMOTE and natural neighbors, Information Sciences,Volume 565,2021,Pages 438-455,

C. Nitin Kumar Mishra, Pramod Kumar Singh,Feature construction and smote-based imbalance handling for multi-label learning,

Information Sciences,563,2021,342-357,

2. The author has utilized Synthetic Minority Oversampling Technique (SMOT). However, there is no prior key explanation about SMOT which is necessary for the general machine learning audience. Include the SMOT algorithm too.

3. The author should include a section to show the robustness of their proposed system against model underfitting and overfitting to claim the practicality of their model.

Round 2

Reviewer 1 Report

The authors did all the updations, as suggested by reviewers.

so i accept this paper